# Incidence of loiasis clinical manifestations in a rural area of the Republic of Congo: Results from a longitudinal prospective study (the MorLo project)

Jérémy T. Campillo[1]*, Biam-Miveck Pakat-Pambou[2], Bachiratou Sahm[1], Sébastien D.S. Pion[1], Marlhand C. Hemilembolo[3], Elodie Lebredonchel[4], Michel Boussinesq[1], François Missamou[3], Cédric B. Chesnais[1]

**1** TransVIHMI, Université de Montpellier, INSERM Unité, Institut de Recherche pour le Développement (IRD), Montpellier, France, **2** Hôpital de référence du district sanitaire de Sibiti, Lékoumou Department, Sibiti, Republic of Congo, **3** Programme National de Lutte contre l'Onchocercose, Direction de l'Épidémiologie et de la Lutte contre la Maladie, Ministère de la Santé et de la Population, Brazzaville, Republic of Congo, **4** Département de Biochimie, Hôpitaux Universitaires Paris Nord Val de Seine – site Bichat, Assistance Publique des Hôpitaux de Paris, Paris, France

* jeremy.campillo@ird.fr

## Abstract

### Background

Loiasis is endemic in Central Africa. Despite evidence of clinical complications and increased mortality, it remains excluded from the list of neglected tropical diseases. The main manifestations are Calabar swellings (CS), Eyeworm (EW) and non-specific general symptoms such arthralgia and pruritus. We calculated incidence rates for clinical manifestations of loiasis from a 13-month study on clinical manifestations in 991 individuals living in *Loa loa*-endemic areas in the Republic of Congo.

### Methodology

From September 2022 to September 2023, community health workers collected weekly symptoms from cohort participants. Detailed data on symptom duration, intensity, associated pruritus, and impact on sleep were recorded. Laboratory procedures included thick blood smear for *L. loa* microfilaremia measurement, creatininemia measurement and eosinophilia counts. We used multiple failure analysis and frailty models to calculate incidence rates of EW, CS, arthralgia, pruritus and absence from work (AfW) and to analyses factors associated with increased incidence of each symptom. The population-attributable fractions (PAFs) associated with loiasis were also calculated for pruritus, AfW and arthralgia.

### Principal findings

Among the studied manifestations, arthralgia had the highest incidence rate at 555.2 cases per 1000 Person-Year (PY), followed by pruritus (332.3 cases/1000 PY), AfW (298.6/1000 PY), EW (266.4/1000 PY), and CS (213.8/1000 PY). Notably, the incidence

**Data availability statement:** The data that support the findings of this study are openly available in DataSuds repository (IRD, France) at https://doi.org/10.23708/BNMZN8. Data reuse is granted under CC-BY license.

**Funding:** This study was fully supported by the European Research Council (ERC) under the European Union's Horizon 2020 research and innovation program (grant agreement No 949963, grant recipient: CBC). The funders had no role in study design, data collection and analysis, decision to publish, or preparation of the manuscript.

**Competing interests:** The authors have declared that no competing interests exist.

rates of CS, pruritus, arthralgia, and AfW were statistically significantly higher in the subgroup of individuals who experienced at least one episode of EW during the follow-up period. EW occurrence is more frequent when microfilaremia is present. The PAFs of AfW, pruritus and arthralgia, associated with loiasis was 18.0% [07.3–27.6], 20.8% [11.6–29.1] and 12.1% [3.1–20.1], respectively.

## Conclusion/significance

This is the first study to provide incidence rates for the clinical manifestations of loiasis. These estimates are crucial for assessing the burden of loiasis. The findings highlight the disease's impact on quality of life.

## Author summary

Loiasis, endemic in Central Africa, presents with clinical complications including Calabar swellings (CS) which are transient angioedemas, Eyeworm (EW) which is the migration of adult worms under the eye conjunctiva, arthralgia, and pruritus. In a 13-month study of 991 individuals in loiasis-endemic areas, arthralgia had the highest incidence rate (555.2 cases per 1000 Person-Years), followed by pruritus, absences, EW, and CS. Notably, rates were elevated in those who experienced EW episodes. EW occurrence is more frequent in individuals with microfilaremia. Absences from work are more frequent in those with a significant history of EW or arthralgia. CS occurrences are more frequent in individuals with a significant history of pruritus, arthralgia, or EW.

## Introduction

Loiasis, caused by the parasitic vector-borne nematode *Loa loa*, is a disease endemic to Central Africa, with over 20.1 million people residing in high-risk regions [1]. Historically, research has primarily addressed challenges related to the co-endemicity of loiasis with other filarial infections (onchocerciasis and lymphatic filariasis), rather than focusing on the health impact of loiasis in itself. Despite substantial evidence suggesting a wide spectrum of clinical complications and increased mortality associated with loiasis, this disease is still not included in the World Health Organization's list of neglected tropical diseases [2–7]. There are currently no specific control programs or recommendations for individual or community treatment of loiasis in endemic countries. Besides serious clinical complications which need further clarification, loiasis also causes well known manifestations that impact the quality of life of affected individuals. The most typical are "Calabar swellings" (CS), which are transient angioedemas, and the migration of adult *L. loa* under the eye conjunctiva, often referred to as "eye worm" (EW). Regarding CS, about 50% of infected adults report having experienced this manifestation at least once in their lifetime. CS can manifest with localized or general pruritus and varying levels of pain [1,8]. EW episodes are transient, lasting from a few hours up to 7 days, are associated with photophobia and ocular itching, inflammation, and pain, usually resolving without complications. About 80% of individuals infected with *L. loa* have already experienced an episode of EW [1,8]. Other unspecific general manifestations include asthenia, headaches, transient nerve palsies, arthralgia, myalgia, pruritus and skin rashes [1,9,10]. It is generally reported that these general symptoms are well known and attributed to loiasis in affected populations [11]. Loiasis is described as a major reason for medical consultations

in highly endemic regions, but there is actually limited data on the prevalence of loiasis-associated manifestations, especially concerning those which are non-specific. Indeed, it is hard to attribute the latter solely to *L. loa* infection because they can be due to other causes (e.g., arthralgia and myalgia can be due to fieldwork or pruritus to other parasitic infections). The article reports the results of the first prospective longitudinal study assessing the incidence and frequency of loiasis-associated common manifestations. To obtain accurate data, a cohort of 991 adults living in *L. loa*-endemic areas in the Republic of Congo was followed up weekly during 13 months.

## Methods

### Ethics statement

The MorLo (Morbidity due to Loiasis) project is an international collaborative study aimed at assessing the prevalence and incidence of *Loa*-related organ-specific complications in rural African areas where loiasis is endemic. This study was approved by the Ethics Committee of the Congolese Foundation for Medical Research (N˚ 036/CIE/FCRM/2022) and by the Congolese Ministry of Health and Population (N˚ 376/MSP/CAB/UCPP-21). All participants received clear and appropriate information and signed an informed consent form for the study.

### Study area and population

In 2022, a cohort including 991 individuals aged ≥18 years old (yo) living in 21 villages within a 50-kilometer radius around Sibiti, the capital town of the Lékoumou Division, was initiated. This region was selected because it is endemic for loiasis, hypoendemic for onchocerciasis, and non-endemic for schistosomiasis and lymphatic filariasis. To date, no mass treatment with ivermectin for onchocerciasis has been implemented, but deworming campaigns with albendazole, restricted to children, are regularly carried out to control soil-transmitted helminthiases. Participants had previously been examined for *L. loa* microfilaremia in 2019, as part of a clinical trial. In 2022, individuals with more than 500 *L. loa* microfilariae (mfs) per mL of blood in 2019 were matched for sex and age (± 5 years) with two individuals living in the same village and identified as amicrofilaremic in 2019.

### Data collection

Initially, community health workers (CHWs) of the 21 villages were trained to identify and record the signs and symptoms possibly associated with loiasis: EW, CS, pruritus, arthralgia and absence from work (including farm work). Pictures of EW (from Tyagi *et al*. [12]) and CS (from Veletzky *et al*. [8]) were distributed to each agent. From 1 September 2022 to 30 September 2023, CHWs visited each cohort participant weekly at home to collect information on events that may have occurred during the preceding week. This was done using a standardized questionnaire (S1 and S2 Files) filled out with the help of the participant. For each manifestation, CHWs collected the start and end dates, the intensity, the presence/absence of associated pruritus (for EW and CS), the presence/absence of associated pain and its frequency (for EW, CS and arthralgia), the impact of the event on sleep, and whether a treatment had been taken. CHWs were asked to report only isolated episode of pruritus as an event or to report an "associated pruritus" if the main event was EW, CS or arthralgia. The filled questionnaires were collected monthly by the central team based in Sibiti and managed using REDCap electronic data capture tools hosted at the Institut de Recherche pour le Développement (Montpellier, France) [13].

## Laboratory procedures

In May-June 2022, 50 μL of blood was collected by finger prick from each participant and spread on a microscope slide to prepare a thick blood smear (TBS) between 10 am and 4 pm to account for diurnal periodicity of the *L. loa* microfilaremia. The slides were dried at room temperature, dehemoglobinized and stained with Giemsa stain within 4 hours. All TBS were examined using a microscope at 100× magnification by experienced technicians to count the *L. loa* mfs. Each TBS was read twice (by two different microscopists), and the arithmetic mean of the counts was used for the statistical analysis. In case of statistically significant discrepancy between the results of the two readings, the slide was read a third time (by one of the two microscopists) and the two closest results were averaged.

Eosinophilia and creatininemia was measured from blood collected in an EDTA or heparinized tube using the HemoCue WBC DIFF System (WBC Diff, HemoCue France, Serris, France) and I-STAT (Abbott Laboratories, Chicago, IL, USA), respectively.

The participant's past exposure to *Onchocerca volvulus* was assessed using an Ov16 Rapid Diagnostic Test (Biplex *L. loa*/Ov16 RDT; Drugs & Diagnostics for Tropical Diseases, San Diego, California). The Ov16 RDT detects antibodies to Ov16 antigen. Two skin snips were collected from each patient with positive Ov16 RDT using a 2 mm Holth-type corneoscleral punch and incubated in saline at room temperature for 24 hours. Emerged mfs were counted under a microscope, and the individuals' *O. volvulus* MFD, expressed as mfs per snip, were calculated using the arithmetic mean of the two snips. Finally, the participants were offered the option to provide stool samples for STH screening. STH infections were identified through the microscopic examination of stool specimens. Participants were supplied with a 50-mL plastic stool container and instructed to collect a morning stool sample. The collected specimens were placed in cooling boxes and shipped to the laboratory within 6 hours. Upon arrival, the samples were either immediately processed or stored overnight at 6°C. Using the Kato-Katz method, a thick smear was prepared from each stool sample and these smears were examined under a microscope at 40× magnification.

## Statistical analysis

Five outcomes were analyzed: occurrence of EW, of CS, of arthralgia, of pruritus and of absence from work (AfW). Analyses were conducted on the number of individuals who experienced one of these events, the number of events over the follow-up period, and their main characteristics (duration, associated pain, frequency of pain, sleep disturbance, associated pruritus and intensity of pruritus, and reason for AfW).

As the events could have occurred multiple times in the same subject during the follow-up period, the data was analyzed using multiple failure analysis. Total incidence rates were calculated for each event. Since EW can be considered a reasonably pathognomonic sign of *L. loa* infection, stratified incidence rates were calculated for the four following groups: microfilaremia > 0 and ≥ 1 episode of EW during the follow-up (MF+/EW+), microfilaremia = 0 and ≥ 1 episode of EW during the follow-up (MF-/EW+), microfilaremia > 0 and no episode of EW during the follow-up (MF+/EW-), and microfilaremia = 0 and no episode of EW during the follow-up (MF-/EW-). All incidence rates were calculated for 1000 person-years (PY). For EW, stratified incidence rates were calculated for the two following groups: microfilaremia > 0 and microfilaremia = 0.

When recurrent events occur, failure times are correlated within the subject and methods that account for the lack of independence are required [14]. We assumed that each event was dependent on the preceding event and used frailty models to introduce this notion with a random covariate [15]. To account for the fact that individuals were not at risk of a new event

until they had recovered from the previous one, we used discontinuous risk intervals [16]. To obtain reliable estimates and avoid small risk-sets, events that occurred after the fourth were disregarded [17]. For each model (when applicable), we used the following covariates: sex, age (categorized as follows: 18–30; 31–40; 41–55; 56–70; > 70 yo), microfilaremia (0; 1–7,999; 8,000–20,000, > 20,000 mfs/mL), eosinophilia (< 0.5, 0.5–1.5 and > 1.5 giga (x10$^9$) cells per liter (G/L)), creatininemia (μmol/L; in continuous), number of EW episodes in the follow-up (0; 1; 2; ≥ 3), number of CS episodes in the follow-up (0; 1; 2; ≥ 3), number of arthralgia episodes in the follow-up (0; 1; 2; ≥ 3), number of pruritus episodes in the follow-up (0; 1; 2; ≥ 3) and STH infection (presence; absence; missing data). In order to measure the impact of loiasis on non-specific symptoms (AfW, pruritus, and arthralgia), we calculated the population-attributable fraction (PAF) [18]. This represents the proportion of these symptoms that could potentially be avoided if loiasis was eliminated. The calculation was based on the frailty models, which included all previously mentioned covariates. For the PAFs calculation, we define loiasis infection as "having one or more event of EW during the follow-up period or/and having microfilaremia". Analyses were performed using Stata, version BE18 (StataCorp LP, College Station, TX). The different patient profiles (number of events during follow-up) are presented in S3 File.

## Results

Table 1 summarizes the characteristics of the 5 events among the cohort population. Among the 991 individuals, 185 (18.7%), 171 (17.3%), 356 (35.9%), 232 (23.4%), and 180 (18.2%) reported at least one event of EW, CS, arthralgia, pruritus and AfW, respectively. Among these patients and over the 13 months of follow-up, the mean number of events reported was 1.5, 1.3, 1.7, 1.5, and 1.8, respectively. Fig 1 represents the distribution of the number of each event in the cohort population.

Twenty-two individuals (2.2%) tested positive for Ov16 RDT and none of them had *O. volvulus* mfs in the skin snips. Out of 771 individuals (77.8%) who volunteered for stool examinations, 391 cases tested positive for STH eggs (50.7%). Among these, there were no cases of hookworm infection, 332 cases of *A. lumbricoides* infection (43.1%), and 207 cases of *T. trichiura* infection (26.8%), with 146 cases exhibiting co-infections (18.8%).

Incidence rates from each event are presented in Table 2 and Fig 2. A total of 1066, 1067, 1063, 1065, and 1065 PY (depending on the duration of the events, which corresponds to the time when individuals are not at risk of experiencing a similar event) were used for calculation of incidence rates, respectively for EW, CS, arthralgia, pruritus and AfW. EW incidence rate was estimated at 329.4/1000 PY in the MF+ group and 231.5/1000 PY in the MF- group. Incidence rates by age and sex are shown in S4 File. EW incidence rate was higher in the 56-70 yo category (341.7; 95% Confidence interval [284.6–410.2]) than in the other age categories (< 30 yo: 88.4 [47.5–164.2]; 31–40 yo: 171.0 [118.1–247.7]; 41–55 yo: 294.9 [243.6–357.0]; > 70 yo: 269.0 [183.1–395.1]. EW incidence rates were respectively 309.9 [259.9–369.5], and 240.2 [205.7–280.5] in the female and male groups, respectively.

CS incidence rates were higher in the MF+/EW+ and MF-/EW+ groups than in the MF-/EW- and MF+/EW- groups. They were higher in the two oldest age groups (56–70 yo category: 273.2 [222.7–335.2]; > 70 yo category: 248.1 [166.3–370.2]) than in the other age categories (< 30 yo: 150.3 [93.4–241.8]; 31–40 yo: 189.4 [133.2–269.3]; 41–55 yo: 179.6 [140.6–229.5]). CS incidence rates were respectively 264.8 [218.9–320.3] and 183.1 [153.3–218.7] in the female and male groups, respectively.

The incidence rate of arthralgia was 2.6 times higher in the MF+/EW+ group than in the MF+/EW- group, and 2.7 times higher in the MF-/EW+ group than in the MF-/EW- group.

**Table 1. Characteristics of the 5 events including the manifestations associated with the latter.**

| | EW | CS | Arthralgia | Pruritus | Absence |
|---|---|---|---|---|---|
| **N of individuals with events (%)** | 185 (18.7) | 171 (17.3) | 356 (35.9) | 232 (23.4) | 180 (18.2) |
| **N of events** | 284 | 228 | 590 | 354 | 318 |
| **N of events per individuals** | | | | | |
| Median [IQR] | 4 [1–6] | 5 [1–4] | 5 [3–6] | 4 [3–6] | 2 [2–2] |
| Mean (min, max) | 1.5 (1, 8) | 1.3 (1, 4) | 1.7 (1, 8) | 1.5 (1, 6) | 1.8 (1, 9) |
| **Duration** (days) | | | | | |
| Median [IQR] | 4 [3–6] | 5 [3–6] | 5 [3–6] | 4 [3–6] | 5 [4–7] |
| Mean (min, max) | 4.6 (1, 18) | 4.0 (1, 20) | 3.9 (1, 23) | 4.8 (1, 17) | 5.8 (1, 31) |
| **Associated pain:** N (%) | | | | | |
| Yes | 237 (89.8) | 185 (83.7) | 535 (100) | 269 (80.8) | |
| Low | 108 (45.6) | 58 (31.3) | 66 (12.3) | 110 (40.9) | |
| Intermediate | 61 (25.7) | 61 (33.0) | 258 (48.2) | 84 (31.2) | |
| Important | 68 (28.7) | 66 (35.7) | 211 (39.4) | 75 (27.9) | |
| MD | 20 | 7 | 45 | 21 | |
| **Pain frequency:** N (%) | | | | | |
| Rare | 46 (19.6) | 26 (14.3) | 75 (13.8) | | |
| Often | 153 (65.1) | 111 (61.0) | 291 (53.5) | | |
| Always | 36 (15.3) | 45 (24.7) | 178 (32.7) | | |
| MD | 2 | 3 | 46 | | |
| **Sleep disturbance:** N (%) | 150 (57.9) | 107 (49.3) | 316 (57.8) | 196 (56.6) | |
| MD | 25 | 11 | 53 | 8 | |
| **Associated pruritus:** N (%) | | | | | |
| Yes | 204 (77.0) | 144 (66.0) | | | |
| Low | 163 (79.9) | 110 (76.4) | | | |
| Important | 41 (20.1) | 34 (23.6) | | | |
| MD | 19 | 10 | | | |
| **Isolated pruritus intensity:** N (%) | | | | | |
| Low | | | | 101 (30.1) | |
| Intermediate | | | | 104 (31.0) | |
| Important | | | | 130 (38.9) | |
| MD | | | | 11 | |
| **Treatment:** N (%) | 61 (23.2) | 67 (30.6) | 237 (42.3) | 102 (29.4) | |
| MD | 21 | 9 | 30 | 7 | |
| **Reason for absence:** N (%) | | | | | |
| Pain | | | | | 169 (56.5) |
| Family reason | | | | | 2 (0.7) |
| Fatigue | | | | | 54 (18.1) |
| Illness | | | | | 73 (24.4) |
| Aging | | | | | 1 (0.3) |
| MD | | | | | 19 |

Abbreviations: N, number; IQR, interquartile range; N/A, not applicable; EW, eyeworm; CS, Calabar swelling; MD, missing data.

Missing data are excluded from percentage calculations.

Arthralgia incidence rate was also higher in the 56–70 yo category (719.3 [634.0–816.1]) than in the other age categories (< 30 yo: 203.4 [135.2–306.1]; 31–40 yo: 330.4 [253.0–431.3]; 41–55 yo: 588.8 [514.1–674.3]; > 70 yo: 655.5 [512.1–839.1]. Arthralgia incidence rates were respectively 664.7 [589.3–749.7] and 489.5 [439.1–545.7] in the female and male groups, respectively.

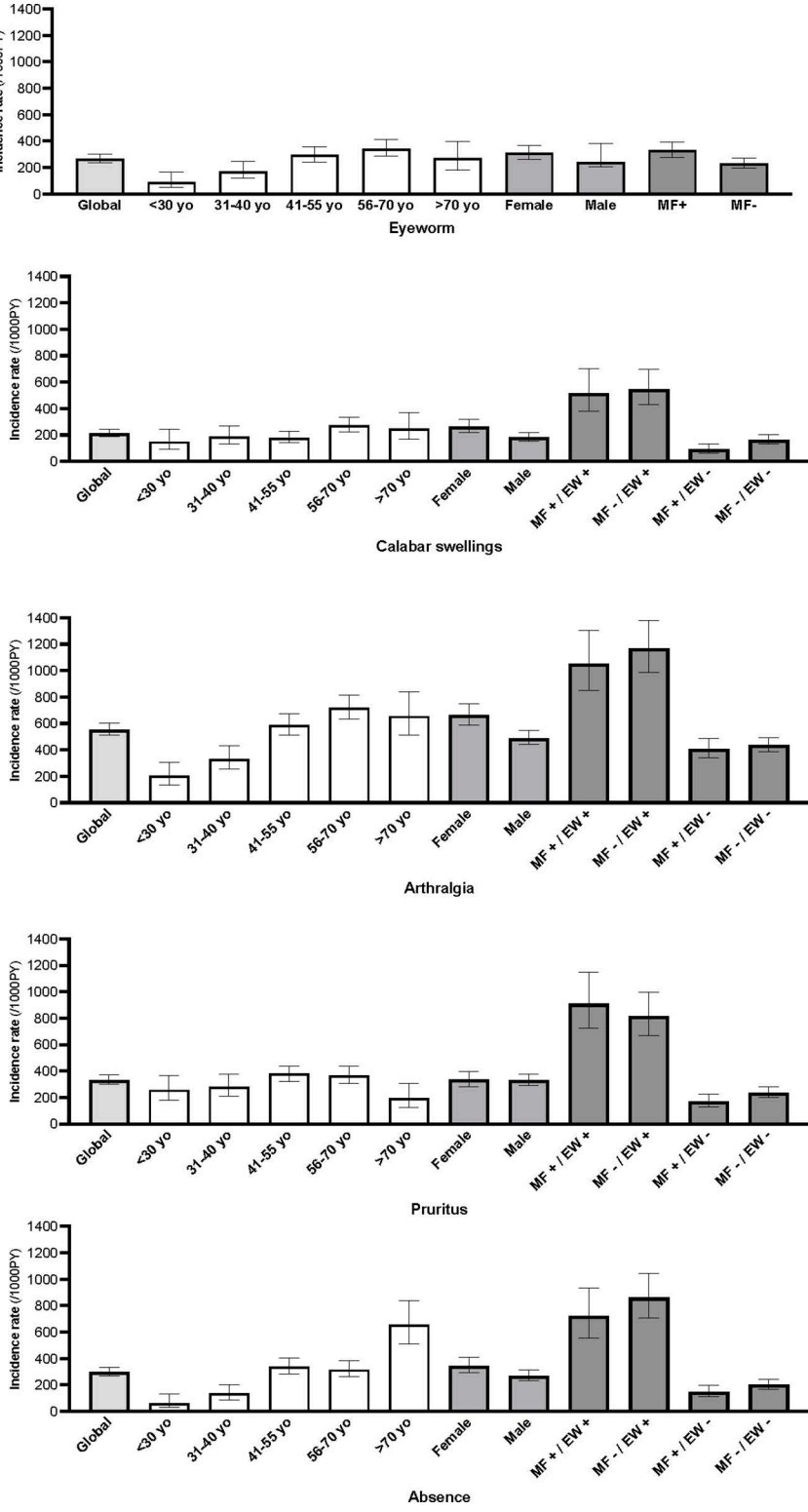

**Fig 1. Frequency of the 5 manifestations in the cohort population.**

Table 2. Incidence rates of the 5 symptoms according to the individual microfilaremia status and the EW status.

| Symptoms | Incidence rate (/1000 PY) | 95%CI |
|---|---|---|
| EW | 266.4 | 237.1– 299.2 |
| MF + | 329.4 | 276.5 – 392.6 |
| MF - | 231.5 | 198.2 – 270.4 |
| CS | 213.8 | 187.8– 243.4 |
| *Loa*-infected* | 268.5 | 226.7 – 318.0 |
| MF +/ EW + | 516.2 | 380.1 – 701.1 |
| MF -/ EW + | 546.2 | 428.3 – 696.5 |
| MF +/ EW - | 93.1 | 64.3 – 134.9 |
| MF -/ EW - | 165.6 | 135.3 – 202.8 |
| Arthralgia | 555.2 | 512.2 – 601.9 |
| *Loa*-infected* | 690.4 | 621.1 – 767.5 |
| MF +/ EW + | 1051.6 | 848.1 – 1304.0 |
| MF -/ EW + | 1167.2 | 987.9 – 1379.2 |
| MF +/ EW - | 407.2 | 340.9 – 486.2 |
| MF -/ EW - | 436.5 | 385.3 – 494.5 |
| Pruritus | 332.3 | 299.4 – 368.8 |
| *Loa*-infected* | 441.7 | 387.0 – 504.1 |
| MF +/ EW + | 909.9 | 722.2 – 1146.3 |
| MF -/ EW + | 817.8 | 670.2 – 997.8 |
| MF +/ EW - | 168.8 | 129.1 – 223.4 |
| MF -/ EW - | 236.2 | 199.4 – 279.8 |
| Absence from work | 298.6 | 267.5 – 333.3 |
| *Loa*-infected* | 407.7 | 355.3 – 467.8 |
| MF +/ EW + | 721.0 | 556.2 – 934.7 |
| MF -/ EW + | 860.7 | 708.9 – 1045.1 |
| MF +/ EW - | 146.5 | 109.0 – 196.9 |
| MF -/ EW - | 202.8 | 169.0 – 243.5 |

Abbreviations: MF; microfilaremia, PY; person-year; CI; confidence intervals; EW, eyeworm; CS, Calabar swellings.

*The *Loa*-infected group is composed of the 3 following categories: MF+/ EW+, MF-/ EW+ and MF+/ EW-.

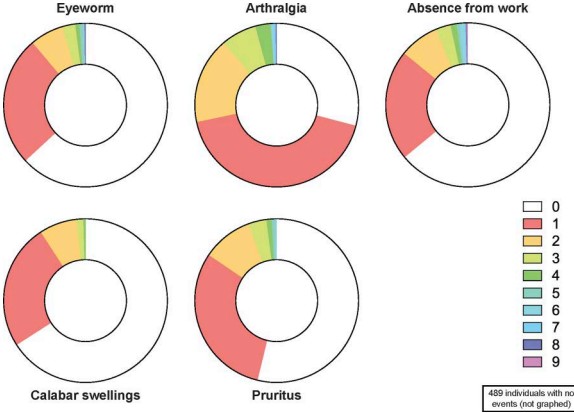

**Fig 2. Incidence of the 5 manifestations by age, sex, microfilaremia and occurrence of an eye worm.** PY, person-year; yo, years old; MF, microfilaremia; EW, eyeworm.

The incidence rate of pruritus was 5.4 times higher in the MF+/EW+ group than in the MF+/EW- group and 3.5 times higher in the MF-/EW+ group than in the MF-/EW- group. Pruritus incidence rates were higher in the 41–55 yo (382.4 [323.2–439.6]) and 56–70 yo categories (368.6 [309.1–439.6]) than in the other age categories (< 30 yo: 256.9 [178.5–368.6]; 31–40 yo: 281.2 [210.6–375.5]; > 70 yo: 196.3 [125.2–307.7]). Pruritus incidence rates were similar in the female (335.0 [282.8–396.8]) and in the male (330.7 [289.8–377.4]) groups.

The incidence rate of AfW was 4.9 times higher in the MF+/EW+ group than in the MF+/EW- group and 4.2 times higher in the MF-EW+ group than in the MF-/EW- group. The incidence rate of AfW was higher in the > 70 yo category (655.0 [511.7–838.5]) than in the other age categories (< 30 yo: 61.9 [29.4–129.8]; 31–40 yo: 134.4 [88.5–204.1]; 41–55 yo: 337.3 [282.1–403.4]; 56–70 yo: 315.4 [260.7–381.5]. Absence incidence rates were respectively 345.3 [292.2–407.9] and 270.6 [233.8–313.2] in the female and male groups, respectively.

Results from frailty models for EW, CS, arthralgia, pruritus and AfW are presented in Tables 3, 4, 5, 6, and 7, respectively. Sex was not associated with the incidence of any of the events. Subjects aged 41–55, 56–70 and > 70 yo were more likely to experience an EW or an arthralgia episode, compared to 18–30 yo individuals. Incidence of CS or pruritus episodes were not associated with age. AfW was more likely to occur in individuals aged 41–55 and > 70 yo than in those aged 18–30 yo. Microfilaremia was associated with occurrence of EW episodes: compared to amicrofilaremic individuals, those with 1–7,999 mfs/mL and 8,000–20,000 mfs/mL were 1.52 (P = 0.004) and 1.71 (P = 0.012) more likely to have experienced an EW episode. In contrast, microfilaremia was not associated with the occurrence of CS, pruritus, arthralgia or AfW episodes. Eosinophilia, creatininemia and STH infections were not associated with the incidence of any of the events.

The number of EW episodes was associated with the occurrence of CS, pruritus, and AfW episodes.

The number of CS episodes was associated with the occurrence of EW, arthralgia, pruritus and AfW episodes. Compared to individuals reporting no CS episode, those who had reported 1 episode were more likely to have experienced an EW, an arthralgia or a pruritus episode, and those reporting 2 episodes were more likely to have had an EW or an AfW episode.

The number of pruritus episodes during the follow–up was associated with the occurrence of an EW episode, with a gradient effect (Hazard Ratio [HR] = 1.82, P < 0.001; HR = 1.92, P = 0.004; and HR = 3.81, P < 0.001, respectively, for 1, 2 and ≥ 3 episodes of pruritus). It was also associated with the occurrence of CS or arthralgia episodes, but not with that of AfW episodes.

The number of arthralgia episodes was associated with the occurrence of EW, pruritus, CS and AfW episodes. Individuals reporting one and two episodes or arthralgia were, respectively, 2.09 (P < 0.001), and 1.54 (P = 0.041) more likely to have experienced an EW episode. The number of arthralgia episodes was strongly associated with the occurrence of a CS episode or of an AfW episode: compared to those with no arthralgia episode, individuals who had reported 1, 2 and ≥ 3 such episodes had 2.66, 3.16, and 2.42 more chance to have had a CS episode, respectively (P < 0.001 in all cases), and 4.81, 4.70, and 5.50 more chance to have had an AfW episode, respectively (P < 0.001 in all cases).

The PAFs of AfW, pruritus and arthralgia, associated with loiasis was 18.0% [07.3–27.6], 20.8% [11.6–29.1] and 12.1% [3.1–20.1], respectively.

## Discussion

This study allowed us for the first time to prospectively count the incidence rates of the main clinical manifestations of loiasis in an endemic area and to evaluate the factors contributing to their occurrence.

**Table 3.  Eyeworm occurrence: results from frailty model for recurrent events.**

| Eyeworm | Hazard Ratio | 95% CI | P-Value |
|---|---|---|---|
| Sex | | | |
| Female | Ref. | | |
| Male | 1.11 | 0.83 – 1.49 | 0.483 |
| Age (yo) | | | |
| 18–30 | Ref. | | |
| 31–40 | 1.53 | 0.73 – 3.19 | 0.262 |
| 41–55 | 2.13 | 1.09 – 4.18 | 0.028 |
| 56–70 | 2.48 | 1.26 – 4.88 | 0.009 |
| > 70 | 2.46 | 1.14 – 5.34 | 0.022 |
| Microfilaremia (mfs/mL) | | | |
| 0 | Ref. | | |
| 1 – 7,999 | 1.52 | 1.14 – 2.02 | 0.004 |
| 8,000 – 20,000 | 1.71 | 1.13 – 2.60 | 0.012 |
| > 20,000 | 0.78 | 0.35 – 1.74 | 0.545 |
| STH infection | | | |
| Negative | Ref. | | |
| Positive | 0.97 | 0.65 – 1.31 | 0.645 |
| Missing Data | 0.81 | 0.46 – 1.61 | 0.649 |
| Eosinophilia (G/L) | | | |
| < 0.5 | Ref. | | |
| 0.5 – 1.5 | 1.02 | 0.76 – 1.38 | 0.886 |
| > 1.5 | 0.92 | 0.65 – 1.31 | 0.645 |
| MD | 0.87 | 0.46 – 1.61 | 0.649 |
| Creatininemia (µmol/L) | 1.00 | 0.99 – 1.01 | 0.794 |
| Episodes of pruritus during the year | | | |
| 0 | Ref. | | |
| 1 | 1.82 | 1.31 – 2.51 | <0.001 |
| 2 | 1.92 | 1.23 – 2.98 | 0.004 |
| ≥ 3 | 3.81 | 2.45 – 5.94 | <0.001 |
| Episodes of CS during the year | | | |
| 0 | Ref. | | |
| 1 | 1.58 | 1.14 – 2.18 | 0.005 |
| 2 | 1.86 | 1.20 – 2.88 | 0.006 |
| ≥ 3 | 1.47 | 0.76 – 2.87 | 0.254 |
| Episodes of arthralgia during the year | | | |
| 0 | Ref. | | |
| 1 | 2.09 | 1.52 – 2.88 | <0.001 |
| 2 | 1.54 | 1.02 – 2.33 | 0.041 |
| ≥ 3 | 1.40 | 0.93 – 2.12 | 0.111 |

Abbreviations: yo, years old; MD, missing data; CS, Calabar swellings; STH, soil-transmitted helminths.

The median durations of EW and CS episodes were estimated at 4 [IQR, 1–6] and 5 [1–6] days, respectively. This aligns with older data, which estimated the duration of EW episode at between a few hours and 7 days [1,8]. Of the five events of interest, the most reported was arthralgia, followed by pruritus, EW, AfW and CS. The literature reports a prevalence of around 42.2% for EW and 36.0% for CS in the total population (aged ≥ 1 yo) of a highly

**Table 4. Calabar swellings occurrence: results from frailty model for recurrent events.**

| Calabar swellings | Hazard Ratio | 95% CI | P-Value |
|---|---|---|---|
| Sex | | | |
| Female | Ref. | | |
| Male | 0.93 | 0.67 – 1.28 | 0.655 |
| Age (yo) | | | |
| 18–30 | Ref. | | |
| 31–40 | 1.18 | 0.59 – 1.98 | 0.814 |
| 41–55 | 0.75 | 0.43 – 1.32 | 0.322 |
| 56–70 | 0.86 | 0.49 – 1.52 | 0.613 |
| > 70 | 1.33 | 0.68 – 2.61 | 0.404 |
| Microfilaremia (mfs/mL) | | | |
| 0 | Ref. | | |
| 1 – 7,999 | 0.73 | 0.51 – 1.04 | 0.081 |
| 8,000 – 20,000 | 1.09 | 0.64 – 1.83 | 0.756 |
| > 20,000 | 1.12 | 0.51 – 2.69 | 0.714 |
| STH infection | | | |
| Negative | Ref. | | |
| Positive | 0.78 | 0.58 – 1.05 | 0.104 |
| Missing Data | 0.73 | 0.49 – 1.09 | 0.129 |
| Eosinophilia (G/L) | | | |
| < 0.5 | Ref. | | |
| 0.5 – 1.5 | 0.81 | 0.58 – 1.13 | 0.221 |
| > 1.5 | 0.88 | 0.61 – 1.29 | 0.525 |
| MD | 0.73 | 0.40 – 1.27 | 0.251 |
| Creatininemia (μmol/L) | 1.00 | 0.99 – 1.01 | 0.550 |
| Episodes of pruritus during the year | | | |
| 0 | Ref. | | |
| 1 | 1.68 | 1.19 – 2.37 | 0.003 |
| 2 | 1.75 | 1.11 – 2.76 | 0.016 |
| ≥ 3 | 2.09 | 1.20 – 3.65 | 0.010 |
| Episodes of EW during the year | | | |
| 0 | Ref. | | |
| 1 | 1.76 | 1.23 – 2.47 | 0.002 |
| 2 | 1.96 | 1.20 – 3.19 | 0.007 |
| ≥ 3 | 1.82 | 1.06 – 3.11 | 0.029 |
| Episodes of arthralgia during the year | | | |
| 0 | Ref. | | |
| 1 | 2.66 | 1.86 – 3.81 | <0.001 |
| 2 | 3.16 | 2.06 – 4.84 | <0.001 |
| ≥ 3 | 2.42 | 1.48 – 3.95 | <0.001 |

Abbreviations: yo, years old; MD, missing data; EW, Eyeworm; STH, soil-transmitted helminths.

*Loa*-endemic area of rural Gabon [8]. Here, we report lower prevalence rates in an adult population of the Congo, with 185 out of 991 individuals (18.7%) having had an EW episode and 171 out of 991 (17.3%) having reported a CS episode. Some individuals experienced up to 8 EW episodes and 4 CS episodes during the 13-month follow-up. Associated pain was reported more frequently (and with greater intensity) in the case of arthralgia, followed by CS, pruritus

**Table 5. Arthralgia occurrence: results from frailty model for recurrent events.**

| Arthralgia | Hazard Ratio | 95% CI | P-Value |
|---|---|---|---|
| Sex | | | |
| Female | Ref. | | |
| Male | 0.96 | 0.79 – 1.16 | 0.652 |
| Age (yo) | | | |
| 18–30 | Ref. | | |
| 31–40 | 1.33 | 0.81 – 2.18 | 0.257 |
| 41–55 | 1.82 | 1.17 – 2.83 | 0.008 |
| 56–70 | 1.88 | 1.20 – 2.92 | 0.005 |
| > 70 | 2.02 | 1.22 – 3.45 | 0.006 |
| Microfilaremia (mfs/mL) | | | |
| 0 | Ref. | | |
| 1 – 7,999 | 1.02 | 0.83 – 1.24 | 0.856 |
| 8,000 – 20,000 | 0.98 | 0.69 – 1.38 | 0.899 |
| > 20,000 | 0.93 | 0.54 – 1.60 | 0.780 |
| STH infection | | | |
| Negative | Ref. | | |
| Positive | 0.96 | 0.80 – 1.15 | 0.651 |
| Missing Data | 0.79 | 0.61 – 1.02 | 0.066 |
| Eosinophilia (G/L) | | | |
| < 0.5 | Ref. | | |
| 0.5 – 1.5 | 0.93 | 0.76 – 1.14 | 0.473 |
| > 1.5 | 0.81 | 0.64 – 1.03 | 0.084 |
| MD | 0.78 | 0.53 – 1.14 | 0.199 |
| Creatininemia (µmol/L) | 1.00 | 0.99 – 1.00 | 0.525 |
| Episodes of pruritus during the year | | | |
| 0 | Ref. | | |
| 1 | 1.60 | 1.30 – 1.98 | <0.001 |
| 2 | 1.58 | 1.18 – 2.13 | 0.002 |
| ≥ 3 | 1.75 | 1.22 – 2.50 | <0.001 |
| Episodes of CS during the year | | | |
| 0 | Ref. | | |
| 1 | 1.53 | 1.23 – 1.89 | <0.001 |
| 2 | 1.37 | 0.97 – 1.94 | 0.077 |
| ≥ 3 | 1.53 | 0.94 – 2.50 | 0.087 |
| Episodes of EW during the year | | | |
| 0 | Ref. | | |
| 1 | 1.43 | 1.15 – 1.78 | 0.001 |
| 2 | 1.30 | 0.92 – 1.81 | 0.138 |
| ≥ 3 | 1.10 | 0.74 – 1.64 | 0.630 |

Abbreviations: yo, years old; MD, missing data; CS, Calabar swellings; EW, Eyeworm; STH, soil-transmitted helminths.

and EW. In more than half of the cases of each manifestation, the events were associated with pain and caused sleep impairment. EW and CS were associated with pruritus in 77.0% and 66.0% of the cases, respectively, and this pruritus was considered severe in approximately 20% of cases. Isolated pruritus was considered important in 40% of cases. However, it's important

**Table 6. Pruritus occurrence: results from frailty model for recurrent events.**

| Pruritus | Hazard Ratio | 95% CI | P-Value |
|---|---|---|---|
| Sex | | | |
| Female | Ref. | | |
| Male | 1.22 | 0.94 – 1.59 | 0.130 |
| Age (yo) | | | |
| 18–30 | Ref. | | |
| 31–40 | 1.00 | 0.62 – 1.61 | 0.988 |
| 41–55 | 0.85 | 0.55 – 1.31 | 0.467 |
| 56–70 | 0.84 | 0.54 – 1.30 | 0.428 |
| > 70 | 0.54 | 0.29 – 1.00 | 0.051 |
| Microfilaremia (mfs/mL) | | | |
| 0 | Ref. | | |
| 1 – 7,999 | 0.90 | 0.69 – 1.17 | 0.446 |
| 8,000 – 20,000 | 0.82 | 0.51 – 1.29 | 0.386 |
| > 20,000 | 1.36 | 0.74 – 2.47 | 0.320 |
| STH infection | | | |
| Negative | Ref. | | |
| Positive | 1.11 | 0.87 – 1.41 | 0.403 |
| Missing Data | 0.87 | 0.63 – 1.20 | 0.405 |
| Eosinophilia (G/L) | | | |
| < 0.5 | Ref. | | |
| 0.5 – 1.5 | 0.90 | 0.69 – 1.18 | 0.446 |
| > 1.5 | 0.86 | 0.62 – 1.18 | 0.344 |
| MD | 1.21 | 0.78 – 1.86 | 0.394 |
| Creatininemia (μmol/L) | 1.00 | 0.99 – 1.00 | 0.563 |
| Episodes of CS during the year | | | |
| 0 | Ref. | | |
| 1 | 1.31 | 0.99 – 1.74 | 0.062 |
| 2 | 1.36 | 0.90 – 2.07 | 0.142 |
| ≥ 3 | 1.57 | 0.88 – 2.78 | 0.123 |
| Episodes of arthralgia during the year | | | |
| 0 | Ref. | | |
| 1 | 1.96 | 1.47 – 2.62 | <0.001 |
| 2 | 2.30 | 1.62 – 3.33 | <0.001 |
| ≥ 3 | 2.68 | 1.86 – 3.86 | <0.001 |
| Episodes of EW during the year | | | |
| 0 | Ref. | | |
| 1 | 1.70 | 1.27 – 2.26 | <0.001 |
| 2 | 2.13 | 1.43 – 3.17 | <0.001 |
| ≥ 3 | 2.37 | 1.52 – 3.70 | <0.001 |

Abbreviations: yo, years old; MD, missing data; CS, Calabar swellings; EW, Eyeworm; STH, soil-transmitted helminths.

to note that data related to the intensity or frequency of pain and pruritus is subjective and can vary greatly from individual to individual.

Our study found lower prevalence rates of EW and CS in the adult population of the Congo (18.7% and 17.3%, respectively) compared to the higher prevalence rates reported in

**Table 7. Absence occurrence: results from frailty model for recurrent events.**

| Absence | Hazard Ratio | 95% CI | P-Value |
|---|---|---|---|
| Sex | | | |
| Female | Ref. | | |
| Male | 1.27 | 0.96 – 1.69 | 0.096 |
| Age (yo) | | | |
| 18–30 | Ref. | | |
| 31–40 | 1.85 | 0.78 – 4.38 | 0.161 |
| 41–55 | 2.47 | 1.12 – 5.43 | 0.025 |
| 56–70 | 2.03 | 0.92 – 4.50 | 0.081 |
| > 70 | 4.05 | 1.78 – 9.18 | 0.001 |
| Microfilaremia (mfs/mL) | | | |
| 0 | Ref. | | |
| 1 – 7,999 | 0.83 | 0.62 – 1.10 | 0.192 |
| 8,000 – 20,000 | 0.67 | 0.49 – 1.15 | 0.147 |
| > 20,000 | 1.30 | 0.67 – 2.51 | 0.443 |
| STH infection | | | |
| Negative | Ref. | | |
| Positive | 1.22 | 0.93 – 1.60 | 0.154 |
| Missing Data | 1.36 | 0.97 – 1.91 | 0.070 |
| Eosinophilia (G/L) | | | |
| < 0.5 | Ref. | | |
| 0.5 – 1.5 | 0.93 | 0.69 – 1.24 | 0.607 |
| > 1.5 | 1.11 | 0.79 – 1.57 | 0.542 |
| MD | 0.68 | 0.35 – 1.31 | 0.253 |
| Creatininemia (µmol/L) | 0.99 | 0.99 – 1.00 | 0.098 |
| Episodes of pruritus during the year | | | |
| 0 | Ref. | | |
| 1 | 0.95 | 0.70 – 1.30 | 0.755 |
| 2 | 0.68 | 0.42 – 1.11 | 0.120 |
| ≥ 3 | 1.29 | 0.82 – 2.04 | 0.275 |
| Episodes of CS during the year | | | |
| 0 | Ref. | | |
| 1 | 1.21 | 0.89 – 1.63 | 0.218 |
| 2 | 1.59 | 1.00 – 2.52 | 0.050 |
| ≥ 3 | 1.42 | 0.78 – 2.60 | 0.251 |
| Episodes of arthralgia during the year | | | |
| 0 | Ref. | | |
| 1 | 4.81 | 3.39 – 6.83 | <0.001 |
| 2 | 4.70 | 3.06 – 7.21 | <0.001 |
| ≥ 3 | 5.50 | 3.64 – 8.32 | <0.001 |
| Episodes of EW during the year | | | |
| 0 | Ref. | | |
| 1 | 1.50 | 1.11 – 2.04 | 0.009 |
| 2 | 1.65 | 1.06 – 2.56 | 0.027 |
| ≥ 3 | 2.74 | 1.77 – 4.24 | <0.001 |

Abbreviations: yo, years old; MD, missing data; CS, Calabar swellings; EW, Eyeworm; STH, soil-transmitted helminths.

a highly Loa-endemic area of rural Gabon (42.2% for EW and 36.0% for CS) [8]. This discrepancy may be attributed to differences in the endemicity of *Loa loa* or in the demographic characteristics of the study populations.

The frequencies of events in our study for the MF+/EW+, MF-/EW+, MF+/EW-, and MF-/EW- groups were similar to those reported in the Gabon study for CS and arthralgia. Regarding pruritus, we observed a higher incidence rate in the EW+ groups compared to the EW- groups, whereas the Gabon study did not find any significant difference. This difference could be due to variations in the reporting and perception of pruritus among the study populations or differences in the local healthcare practices and diagnostic criteria.

Additionally, our findings confirm the previous observation that patients with a history of EW have a much higher clinical burden than those with microfilaremia only. This supports the classification of "migratory" loiasis (MF+/EW+ and MF-/EW+) versus "non-migratory" loiasis (MF+/EW-) which effectively determines the clinical penetrance of loiasis. This classification could be beneficial in determining the clinical impact of loiasis and should be considered in future studies.

One of the limitations of assessing pruritus is that it can be isolated or associated with other symptoms. The episodes of pruritus reported concerned only isolated pruritus and not pruritus associated with other manifestations. It is therefore likely that the prevalence and incidence of pruritus are underestimated. Therefore, such data should be interpreted with caution. Lastly, treatment was taken in 42% of cases for arthralgia, around 30% for CS and pruritus and 23% for EW. Treatment will be analyzed in a specific article. The majority of AfW (56.5%) were due to pain.

Similar to the prevalence rates, incidence analysis showed that arthralgia was the most common disorder, with 555.2 cases per 1000 PY, followed by pruritus (332.3 cases/1000 PY), absences from work (298.6/1000 PY), EW (266.4/1000 PY), and CS (213.8/1000 PY). These initial incidence data are crucial for assessing the burden of loiasis in populations living in endemic areas.

Regarding specific events, microfilaremia did not appear to statistically significantly impact incidence rates of CS, arthralgia, pruritus and AfW. In contrast, the occurrence of an episode of EW during follow-up was strongly associated with the occurrence of CS, pruritus, and arthralgia episodes. It is important to note that some reported events may not be directly attributable to loiasis. Nevertheless, calculating the difference in incidence rates between the EW+ and EW- groups serves as a useful proxy for estimating the proportion of symptoms genuinely related to *L. loa* infection.

In the multiple failure survival analysis, symptom occurrence did not seem influenced by sex but was clearly influenced by older age. For the first time, we observed that EW occurrence appears to be associated with microfilaremia levels. Significance was not achieved for the >20,000 mfs/mL group, but this may be due to low statistical power or to unknown biological factors. More studies are needed to better understand this phenomenon. Neither eosinophilia nor creatininemia was associated with any of the events studied. The number of EW episodes was associated with the occurrence of CS, pruritus, and AfW episodes.

Similarly, the number of CS episodes was associated with the occurrence of EW, pruritus, and AfW episodes. Individuals reporting one CS episode were more likely to experience arthralgia and pruritus, while those reporting two episodes were more likely to report AfW. The number of pruritus episodes during follow-up showed a gradient effect, with increasing hazard ratios for occurrence of EW, CS, and arthralgia episodes. Arthralgia episodes were strongly associated with the occurrence of CS and AfW episodes. Reporting one or two episodes of arthralgia statistically significantly increased the likelihood of experiencing CS. Individuals reporting arthralgia episodes during follow-up were substantially more likely to report AfW. The results and the observed patterns of event occurrence, as detailed in S3 File, reveal distinct groups among individuals: (i) non-event reporters (489 individuals who never reported any events during the

follow-up period); (ii) single event experiencers (160 participants who experiencing a single type of event, often with low frequencies) (Fig 2); and (iii) a high event frequency group, with some individuals having reported up to 8 events during the follow-up period. Further analysis is warranted to gain deeper insights. Identifying factors that contribute to differential susceptibility to these disorders among patient groups will be crucial for a comprehensive understanding.

The PAFs that were calculated turned out to be quite high, particularly for AfW, which had a population attributable fraction of 18.0%. This suggests that in our study population, approximately 18.0% of absences could potentially be eliminated if loiasis were to be eradicated, assuming all other factors remain constant. The socioeconomic impact of loiasis, including its effect on work absenteeism, underscores the importance of addressing this disease to improve overall community health and productivity. However, when it comes to loiasis, establishing a definitive diagnosis is still not feasible. The diagnosis relies on the presence of microfilaremia, which is hidden in over 40% of cases, or the manifestation of a pathognomonic symptom such as EW or CS. Even though a differential diagnosis is possible for the latter, we have opted to define the "loiasis-affected" population for our PAF calculations as those who have experienced at least one instance of EW and/or CS during the follow-up period. It's important to note that population attributable fractions largely depend on the prevalence of exposure, and in the case of loiasis, the exposure can only be an estimate. Therefore, these fractions should be interpreted with caution and within the context of a specific population.

The strengths of our study include its prospective design with a relatively long follow-up period of 13 months, the inclusion of nearly one thousand individuals, and the close weekly monitoring of the subjects. Additionally, the use of adapted statistical methods for repeated events, which account for high unmeasured heterogeneity and discontinuous risk intervals, further enhances the robustness of our findings.

However, several limitations should be considered. First, there may be underreporting of some events by CHWs due to inter-observer variability in their capacity to prospectively collect information. Fatigue due to long-term data collection could also have impacted the accuracy of symptom reporting over time. Attributing some symptoms to loiasis with absolute certainty is challenging. However, it's quite improbable that EW and, to a smaller degree, CS result from other causes. That being said, pruritus and arthralgia could potentially have other origins. Nevertheless, the substantial difference in incidence rates between the EW+ and EW- groups suggests that the majority of events are, at least partially, attributable to loiasis. Other pathologies could potentially cause pruritus, arthralgia, and edema (CHWs might confuse CS with other types of edema). Given the similar health, demographic, and environmental conditions across villages, any confusion is unlikely to statistically significantly differ among different villages or individuals. The study population was selected according to microfilaremia, with each microfilaremia was matched with two other individuals of the same sex and similar age from the same village. However, since microfilaremia does not appear to be statistically significantly involved in the various health events, it is unlikely that that this selection method has led to an under- or over-estimation of the calculated incidence rates. To our knowledge, this study is the first to prospectively count the incidence rates of the main clinical manifestations of loiasis in an endemic population. These estimates are crucial for understanding the burden of this disease. In addition to the well-documented excess mortality and severe clinical complications associated with loiasis, recent data shed light on its broader implications. Specifically, these findings underscore the impact of loiasis on the quality of life, and consequently, on the socio-economic conditions of the endemic populations. As we delve deeper into the repercussions of loiasis, it becomes evident that it should be recognized as a neglected tropical disease. Thoughtful consideration is also warranted regarding loiasis control strategies in regions where it is prevalent.

## Supporting information

**S1 File. Questionnaire (in French).**
(PDF)

**S2 File. Questionnaire (translated in English).**
(PDF)

**S3 File. Patterns of the 5 symptoms among the population.**
(DOCX)

**S4 File. Incidence rates of the 5 symptoms according to the age and sex.**
(DOCX)

## Acknowledgments

We thank the French Embassy in Republic of Congo. We thank the Lékoumou health district, the medical, paramedical and technical staff of the Sibiti hospital, the PNLO and IRD drivers, and the participants for agreeing to participate. Finally, we would like to pay tribute to the late Mr Mpandzou, for his invaluable help with all the work in the field, his invaluable support with local people and his unwavering commitment to improving the health conditions of its population.

## Author contributions

**Conceptualization:** Michel Boussinesq, Cédric B Chesnais.

**Data curation:** Bachiratou Sahm.

**Formal analysis:** Jérémy T Campillo.

**Funding acquisition:** Cédric B Chesnais.

**Investigation:** Jérémy T Campillo, Biam-Miveck Pakat-Pambou, Marlhand C. Hemilembolo, Elodie Lebredonchel, François Missamou, Cédric B Chesnais.

**Methodology:** Jérémy T Campillo, Cédric B Chesnais.

**Project administration:** Marlhand C. Hemilembolo, François Missamou.

**Resources:** Jérémy T Campillo, Marlhand C. Hemilembolo, François Missamou, Cédric B Chesnais.

**Supervision:** Michel Boussinesq, Cédric B Chesnais.

**Validation:** Jérémy T Campillo, Cédric B Chesnais.

**Visualization:** Jérémy T Campillo.

**Writing – original draft:** Jérémy T Campillo.

**Writing – review & editing:** Jérémy T Campillo, Biam-Miveck Pakat-Pambou, Bachiratou Sahm, Sébastien D S Pion, Marlhand C. Hemilembolo, Elodie Lebredonchel, Michel Boussinesq, François Missamou, Cédric B Chesnais.

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
