## [Decision Letter · Decision Letter 0]

27 Oct 2024

PNTD-D-24-01074Incidence of loiasis clinical manifestations in a rural area of the Republic of Congo: results from a longitudinal prospective study (the MorLo project)PLOS Neglected Tropical Diseases Dear Dr. Campillo, Thank you for submitting your manuscript to PLOS Neglected Tropical Diseases. After careful consideration, we feel that it has merit but does not fully meet PLOS Neglected Tropical Diseases's publication criteria as it currently stands. Therefore, we invite you to submit a revised version of the manuscript that addresses the points raised during the review process. Please submit your revised manuscript within 60 days Dec 26 2024 11:59PM. If you will need more time than this to complete your revisions, please reply to this message or contact the journal office at plosntds@plos.org. Please include the following items when submitting your revised manuscript:* A rebuttal letter that responds to each point raised by the editor and reviewer(s). You should upload this letter as a separate file labeled 'Response to Reviewers '. This file does not need to include responses to any formatting updates and technical items listed in the 'Journal Requirements' section below.* A marked-up copy of your manuscript that highlights changes made to the original version. You should upload this as a separate file labeled 'Revised Manuscript with Track Changes '.* An unmarked version of your revised paper without tracked changes. You should upload this as a separate file labeled 'Manuscript '. If you would like to make changes to your financial disclosure, competing interests statement, or data availability statement, please make these updates within the submission form at the time of resubmission. Guidelines for resubmitting your figure files are available below the reviewer comments at the end of this letter. We look forward to receiving your revised manuscript. Kind regards, Feng Xue, Ph.D.Guest EditorPLOS Neglected Tropical Diseases Jong-Yil ChaiSection EditorPLOS Neglected Tropical Diseases

Shaden Kamhawi

co-Editor-in-Chief

Paul Brindley

co-Editor-in-Chief

**Journal Requirements:** **Additional Editor Comments (if provided):****Reviewers' Comments:** Reviewer's Responses to Questions

**Key Review Criteria Required for Acceptance?**

**Methods**

-Are the objectives of the study clearly articulated with a clear testable hypothesis stated?

-Is the study design appropriate to address the stated objectives?

-Is the population clearly described and appropriate for the hypothesis being tested?

-Is the sample size sufficient to ensure adequate power to address the hypothesis being tested?

-Were correct statistical analysis used to support conclusions?

-Are there concerns about ethical or regulatory requirements being met?

Reviewer #1: (No Response)

Reviewer #2: Yes, objectives of the study clearly are articulated with a clear testable hypothesis stated.

Yes, the study design is appropriate to address the stated objectives.

Yes, the population is clearly described and appropriate for the hypothesis being tested.

I don't not have sufficient statistical expertise to provide an informed critique

Reviewer #3: Methods clear

Reviewer #4: -Are the objectives of the study clearly articulated with a clear testable hypothesis stated?

Yes

-Is the study design appropriate to address the stated objectives?

Yes

-Is the population clearly described and appropriate for the hypothesis being tested?

Yes

-Is the sample size sufficient to ensure adequate power to address the hypothesis being tested?

Yes

-Were correct statistical analysis used to support conclusions?

Yes

-Are there concerns about ethical or regulatory requirements being met?

No

**Results**

-Does the analysis presented match the analysis plan?

-Are the results clearly and completely presented?

-Are the figures (Tables, Images) of sufficient quality for clarity?

Reviewer #1: (No Response)

Reviewer #2: Yes the results are clearly and completely presented?

Yes the figures (Tables, Images) are of sufficient quality for clarity?

Reviewer #3: Results clear

Reviewer #4: -Does the analysis presented match the analysis plan?

Yes

-Are the results clearly and completely presented?

Table 1 is not fully visible on the provided version

-Are the figures (Tables, Images) of sufficient quality for clarity?

see above

**Conclusions**

-Are the conclusions supported by the data presented?

-Are the limitations of analysis clearly described?

-Do the authors discuss how these data can be helpful to advance our understanding of the topic under study?

-Is public health relevance addressed?

Reviewer #1: (No Response)

Reviewer #2: Yes the conclusions are supported by the data presented

Yes the limitations of analysis are clearly described

Yes the authors discusses how these data can be helpful to advance our understanding of the topic under study

Yes the public health relevance is addressed?

Reviewer #3: Conclusion justified

Reviewer #4: -Are the conclusions supported by the data presented?

yes

-Are the limitations of analysis clearly described?

yes

-Do the authors discuss how these data can be helpful to advance our understanding of the topic under study?

see comment, I think the new data can be discussed more with regard to available literature.

-Is public health relevance addressed?

see comment above

**Editorial and Data Presentation Modifications?**

Reviewer #1: (No Response)

Reviewer #2: Minor revision

Reviewer #3: see below

Reviewer #4: (No Response)

**Summary and General Comments**

Reviewer #1: This manuscript examines the incidence of eyeworm, Calabar swellings, arthralgia, and pruritus in a Loa loa-endemic region of Congo as part of a very comprehensive project examining the morbidity associated with loiasis. The authors argue, given the high incidence of these signs/symptoms, that there is a clear impact on the quality of life of those affected.

Using a number of tools and weekly collected data obtained through community health workers, the study team was able to document the incidence of the various Loa-specific and Loa-non-specific clinical manifestations of loiasis and perform associations with microfilaremia, eosinophilia, and renal disease.

The study provides very important information but several issues need to be addressed:

1)Because the regions of study are hypoendemic for onchocerciasis, it is possible that up to 20 percent are co-infected with O. volvulus (Ov). Thus, it would be important to know how Ov infection is contributing to the pruritus and/or arthralgia incidence. Serologic assessment (e.g. antibodies to Ov16) would be extremely helpful. At a minimum, the authors should put into their statistical models a potential confounder of up to 20% Ov prevalence to see if this changes the real incidence of these 2 non-specific clinical manifestations.

2)Similarly, the authors very likely know the soil-transmitted helminth (STH) infection status of the group. Again, as many of the non-specific manifestations could be related to STHs, having this information will be critical.

3)On a similar track, what about scabies, lice, other ectoparasites that are known to cause intense pruritus?

Reviewer #2: In this manuscript entitled “Incidence of loiasis clinical manifestations in a rural area of the Republic of Congo: results from a longitudinal prospective study (the MorLo project)”, the authors provided much-needed information about the incidence rates of diverse clinical manifestations of loiasis. The experimental design and data presentation are clearly and support to the conclusion. The English of this manuscript should be carefully polished. Please consider all issues and concerns mentioned in the comments and suggestions, which hopefully can help authors improve the study. Specifically, please consider the following points before further process:

Abstract

-The aim of the study should be clearly stated in the background section.

-In the background section, page 2 line 33: “L. loa” should be defined the first time they are used.

-In results section, page 2 line 45: should specify if incidence rated was statistically significant rather than just say 'a significant elevated'.

Introduction

-Abbreviations made in the Abstract, should be redefined in the main text the first time they are used.

-It is better to rearrange clinical manifestations with logical order as follows:

1-“The most typical are “Calabar swellings” (CS), which are transient angioedemas, and the migration of adult L. loa under the eye conjunctiva, often referred to as “eye worm” (EW)” (page 4, lines 71-73)

2-“Regarding CS, about 40% of infected adults report having experienced this manifestation at least once in their lifetime. CS can manifest with localized or general pruritus and varying levels of pain” (page 4, lines 84-86)

3-“EW episodes are transient, lasting from a few hours up to 7 days, are associated with photophobia and ocular itching, inflammation, and pain, usually resolving without complications” (page 4, lines 82-84)

4-“Other unspecific general manifestations include asthenia, headaches, transient nerve palsies, arthralgia, myalgia, pruritus and skin rashes [8–10]. It is generally reported that these general symptoms are well known and attributed to loiasis in affected populations [11]. Loiasis is described as a major reason for medical consultations in highly endemic regions, but there is actually limited data on the prevalence of loiasis-associated manifestations, especially concerning those which are non-specific. Indeed, it is hard to attribute the latter solely to L. loa infection because they can be due to other causes (eg. arthralgia and myalgia can be due to fieldwork or pruritus to other parasitic infections).” (page 4, lines 82-84)

Methods

-The ethical approval number was not stated.

-Minor changes:

•Page 5 line 104-107: Detete “To date, no mass treatment with ivermectin for onchocerciasis had been implemented, but deworming campaigns with albendazole, restricted to children, are regularly carried out to control soil-transmitted helminthiases.”

•Page 5 line 101: redefine (yo)

•Page 5 line 101: redefine abbreviation of “microfilaremia” the first time it is used “mfs”

•Page 7 line 144: Don’t repeat “of” just the first time it is used in sentence.

Results

-Cut off point of statistically significant results should be clearly mentioned and stated in all tables (p value).

-Any statistically significant result should be specified “statistically significant” rather than just say “significant”.

-p in statistical analysis italic

Discussion

-The discussion is well organized and written. However, it is better to mention limitations of assessing pruritus “One of the limitations of assessing pruritus is that it can be isolated or associated with other symptoms. The episodes of pruritus reported concerned only isolated pruritus and not pruritus associated with other manifestations. associated with other manifestations. It is therefore likely that the prevalence and incidence of pruritus are underestimated.” (page 21 lines 308-309 and page 22 lines 310-311) “before “Therefore, such data should be interpreted with caution.” (page 21 line 304)

-Recommendations should be stated in conclusion.

Reviewer #3: The authors report on a landmark project (MorLo) that sets out to better define the clinical characteristics of loiasis in patients of a high transmission region in the Republic of Congo. The study was diligently designed, carefully conducted, and the analysis of the incidence of common signs and symptoms described in this paper is highly important. Until today, we mostly rely on data from cross-sectional surveys about the incidence of common signs and symptoms whereas this study allowed for the first time to assess the incidence and duration of these signs and symptoms.

The main findings are a substantial population attributable fraction of absence from work (22%), arthralgia (13%), and pruritus (17%). Importantly, these common symptoms were associated with the history of “Eyeworm Migration”, which substantiates the previously proposed concept of “migratory” loiasis showing more often subjective signs and symptoms of loiasis than “non-migratory loiasis”. In addition, the analysis shows for the first time that migratory loiasis (with its associated clinical symptoms) is age dependent.

The authors are the leading experts in the field of human filariasis today, who have shaped with their work many of our current concepts in the understanding of the disease and this work substantially adds to their important contribution to better define the clinical impact of loiasis. The authors have to be commended for their work and the data are much awaited to be published by the interested scientific community. My comments are mostly concerning the way of data presentation and analysis as I would find a slightly different approach more intuitive for the understanding of the important outcome of this study – while at the same time fully appreciating that the way the authors have chosen to present their findings is justified. I hope that some of my thoughts may be taken up in a revised version to be included in the analysis and depiction of data.

Major comments

My main comment concerns the way the data are depicted and analysed. While there is no question at all about the validity of the data and analysis, my feeling is that a slightly different depiction would make it easier for the reader to understand key biological features of loiasis that have been so well defined by this project:

“EW incidence rate 195 was estimated at 329.4/1000 PY in the MF+ group and 231.5/1000 PY in the MF- group.“ This analysis and the main depiction of data in table 1 and the results section on incidence of signs and symptoms:

To me this sounds as if these are incidence rates for the entire population, which include L. loa infected and uninfected persons (with about half of the population not reporting any sign or symptoms of loiasis). While this is of interest to understand the overall burden of disease in this community, it would be (more) interesting for me to understand the incidence of EW, CS, arthralgia, pruritus, etc in the 1) L. loa infected individuals (to understand better the disease) and separately in the 2) uninfected individuals (to understand the background for unspecific signs and symptoms). When you depict the data for the total population, the incidence is influenced both by the prevalence of the disease in this community and the frequency of occurrence of symptoms in infected individuals. The results therefore have less external validity for other communities with different prevalence/transmission intensities.

To me it would make more sense to categorize persons in infected and uninfected individuals based on microfilaremia and history of EW as the two most specific markers for occult and microfilaremic loiasis (one may argue whether CS is specific enough or not and include or not this in the case definition) – and calculate the incidence of signs and symptoms for these groups separately. This would provide an estimate of the clinical features of loiasis in the infected patients versus the “background” incidence of symptoms (arthralgia, pruritus, absence from work) in the uninfected. These estimates would have more external validity for individual patients in other transmission settings.

In a way the authors have used this approach by analyzing the data according to EW and MF status, defining the categories EW+/MF+, EW-/MF-, etc. What I would suggest, is to add the combined category EW+/MF+, EW+/MF-, MF+/EW- into one category: Loa infected. This will then provide the incidence for infected versus un-infected persons.

Finally, the authors also evaluate the difference in the occurrence of signs and symptoms in patients with or without history of EW (and CS). The authors confirm the previous finding that patients with history of EW have a much higher clinical burden than patients with MF only (Veletzky et al 2022 PLoSNTD Distinct loiasis infection states and associated clinical and hematological manifestations in patients from Gabon – a study that analyzed these concept for the first time based on cross-sectional data and clearly requires to be discussed in this manuscript). The classification of “migratory” versus “non-migratory” loiasis has been proposed for these two categories of patients and it may be worthwhile for the authors to use or discuss this concept of classification which seems to aptly determine the clinical penetrance of loiasis.

“Microfilaremia was associated with occurrence of EW episodes: 241 compared to amicrofilaremic individuals, those with 1–7,999 mfs/mL and 8,000–20,000 242 mfs/mL were 1.50 (P = 0.005) and 1.71 (P = 0.011) more likely to have experienced an EW 243 episode.“

The understanding of this analysis is difficult for me. As it stands it seems to me that uninfected persons (EW-/MF-) were mixed with occult patients (EW+/MF-) in one group and compared to patients with different levels of MF (and therefore all infected individuals). This does not seem informative to me as analysis, as we actually want to understand whether occult loiasis patients have more or fewer episodes of EW than microfilaremic individuals. This is inherently difficult to analyse as we have no marker for occult disease other than the clinical signs (mostly EW). A way out of this conundrum would be to analyze the number of EW events in those with at least one EW in the MF- versus those with at least one EW in the category of varying levels of MF+. In the current version the non-infected persons (which are mixed with the occult patients) seem to skew the group of MF-.

The association of individual signs&symptoms with each other seems to strongly confirm the concept of migratory loiasis being the main driver for clinical disease as opposed to microfilaraemia (Veletzky et al 2022 PLoSNTD as above). It may therefore be worth considering an analysis using the concept of migratory loiasis versus non-migratory loiasis versus non-infected persons.

Minor comments:

The estimates for the epidemiology of loiasis refer to an outdated analysis from 2011. A newer estimate based on these data and the population growth in rural regions of Central Africa estimates 20.1 million people residing in high transmission regions (as opposed to 11 million) as discussed in reference 10.

The authors define loiasis as persons with either eyeworm or Calabar swelling (line 175). I wonder whether microfilaremia was not regarded as criterion (eg microfilaremia without EW and without CS)? In addition, CS seems not so specific for loiasis as it may also be caused other infections (mansonellosis, etc), rheumatic diseases, etc. It would be good if the authors would comment on this consideration and why they chose the definition as such.

Attributable fraction: it is understood that the attributable fraction of absence from work etc relies on the hypothesis that all other factors are equal between infected and un-infected persons. In reality this may not be the case with infected individuals being more likely to suffer from other infectious diseases, potentially lower socioeconomic status, different work envirnoments, etc. This cautionary thought could be mentioned more clearly for this analysis.

Reviewer #4: The manuscript by Campillo et al reports on the incident rates of various loiasis associated symptoms. The data was collected prospectively during a long follow up period in a large patient cohort as part of a larger study.

The prospective data collection and the short intervals of visits by HCWs with the villagers are factors supporting the quality of the data. In general, the manuscript is well written and easy to follow.

The authors provide symptom incidences and their association with respective loiasis manifestations - with very interesting results. It seems that eyeworm occurrence, in contrary to microfilaremia, is well associated with symptom occurrence.

My major comment is that the here reported results from the Congo are aligned with similar findings that have been described in a cohort from Gabon (https://doi.org/10.1371/journal.pntd.0010793) but this is not discussed.

This actually supports the relevance and validity of the here reported data. I therefore suggest that the authors discuss their findings in regard with previous data, with special regard to similarities and also differences as well as possible conclusions that may be drawn from those findings.

Also, the absence of work and socioeconomic impact of loiasis has been described and this could also be addressed to improve the discussion.

Otherwise I have mostly minor comments, which should be addressed before publication and which are outlined below.

Minor:

Abstract line 46: “EW occurrence appears influenced by microfilaremia.” In what way? Adapt for clarity.

Line 58: “Microfilaremia influenced EW occurrence, and CS correlated with EW, pruritus, and absence.“ Influence in what way? Absence of work? Adapt for clarification.

Line 59: “These 59 findings emphasize the impact of loiasis on quality of life.” As the study is not aiming to address the quality of life (by e.g. using a QALY questionnaire) I suggest to rephrase this final sentence.

Line 105: “To date, no mass treatment with 105 ivermectin for onchocerciasis had been implemented,..” I think “…has been… “ would be correct.

Line 173: “This represents the 173 proportion of these symptoms that could potentially be eradicated if loiasis was eliminated.” Maybe “…potentially be avoided if…” is more adequate?

Table 1: in the PDF the table is not fully visible.

-Line for associated pruritus and CS the % should be 76.4 and 23.6 ?

-Line for isolated pruritus intensity the % numbers don’t add up? (29.4+30.3+37.9= 97,6)

Line 286- 288: “This study allowed us to estimate, for the first time, the incidence rates of the main clinical manifestations of loiasis in an endemic area and to assess the factors contributing to their occurrence.” Estimates have been done before but you have primary data and you literally counted the episodes - so I think you should clearly state that and may write “This study allowed us for the first time to prospectively count the incidence rates of the main clinical manifestations of loiasis in an endemic area and to evaluate the factors contributing to their contribute to their occurrence.”

Line 291 : citation 12 reported these eyeworm durations.

Line 304-306: “Lastly, 305 treatment was taken in 42% of cases for arthralgia, around 30% for CS and pruritus and 23% 306 for EW. Treatment will be analyzed in a specific article.“ Is this similar to another article from Gabon (doi: 10.1371/journal.pntd.0012389)? Would be interesting to compare the results.

Line 317 - 319: “Regarding specific events, microfilaremia did not appear to significantly impact incidence rates 318 of CS, arthralgia, pruritus and AfW. In contrast, the occurrence of an episode of EW during 319 follow-up was strongly associated with the occurrence of CS, pruritus, and arthralgia episodes.” I suggest to discuss this in the light with previous reports. See comment above.

Line 359 – 363: The prospective design of this study, with a relatively long (13 months) follow-up period, the inclusion of nearly one thousand individuals, and the close weekly monitoring of the subjects are notable strengths. Additionally, the use of adapted statistical methods for repeated events with high unmeasured heterogeneity and discontinuous risk intervals are some of the strengths of this study.” “Of this study” is used twice, I suggest to rephrase.

Line 378-380: “To our knowledge, this study is the first to report incidence rates of the main clinical manifestations of loiasis in an endemic population.” See comment above.

PLOS authors have the option to publish the peer review history of their article (what does this mean? ). If published, this will include your full peer review and any attached files.

**Do you want your identity to be public for this peer review?** For information about this choice, including consent withdrawal, please see our Privacy Policy .

Reviewer #1: No

Reviewer #2: No

Reviewer #3: No

Reviewer #4: No

---

## [Decision Letter · Decision Letter 1]

25 Jan 2025

Dear Dr Campillo,

We are pleased to inform you that your manuscript 'Incidence of loiasis clinical manifestations in a rural area of the Republic of Congo: results from a longitudinal prospective study (the MorLo project)' has been provisionally accepted for publication in PLOS Neglected Tropical Diseases.

Best regards,

Feng Xue, Ph.D.

Guest Editor

Jong-Yil Chai

Section Editor

Shaden Kamhawi

co-Editor-in-Chief

Paul Brindley

co-Editor-in-Chief

Reviewer's Responses to Questions

**Key Review Criteria Required for Acceptance?**

**Methods**

-Are the objectives of the study clearly articulated with a clear testable hypothesis stated?

-Is the study design appropriate to address the stated objectives?

-Is the population clearly described and appropriate for the hypothesis being tested?

-Is the sample size sufficient to ensure adequate power to address the hypothesis being tested?

-Were correct statistical analysis used to support conclusions?

-Are there concerns about ethical or regulatory requirements being met?

Reviewer #1: (No Response)

Reviewer #2: (No Response)

Reviewer #3: adequate

Reviewer #4: (No Response)

**Results**

-Does the analysis presented match the analysis plan?

-Are the results clearly and completely presented?

-Are the figures (Tables, Images) of sufficient quality for clarity?

Reviewer #1: (No Response)

Reviewer #2: (No Response)

Reviewer #3: adequate

Reviewer #4: (No Response)

**Conclusions**

-Are the conclusions supported by the data presented?

-Are the limitations of analysis clearly described?

-Do the authors discuss how these data can be helpful to advance our understanding of the topic under study?

-Is public health relevance addressed?

Reviewer #1: (No Response)

Reviewer #2: (No Response)

Reviewer #3: adequate

Reviewer #4: (No Response)

**Editorial and Data Presentation Modifications?**

Reviewer #1: (No Response)

Reviewer #2: (No Response)

Reviewer #3: adequate

Reviewer #4: (No Response)

**Summary and General Comments**

Reviewer #1: The authors have addressed quite satisfactorily the issues raised by the various reviewers and modified quite appropriately the manuscript in this revision.

Reviewer #2: (No Response)

Reviewer #3: I thank the quthors for the diligent revision and the update of the manuscript . All my thoughts have been appropriately addressed.

Reviewer #4: The raised comments have been addressed an I recommend an acception of the paper.

PLOS authors have the option to publish the peer review history of their article (what does this mean? ). If published, this will include your full peer review and any attached files.

**Do you want your identity to be public for this peer review?** For information about this choice, including consent withdrawal, please see our Privacy Policy .

Reviewer #1: No

Reviewer #2: No

Reviewer #3: No

Reviewer #4: No

---

## [Editor Report · Acceptance letter]

Dear Dr Campillo,

We are delighted to inform you that your manuscript, "Incidence of loiasis clinical manifestations in a rural area of the Republic of Congo: results from a longitudinal prospective study (the MorLo project)," has been formally accepted for publication in PLOS Neglected Tropical Diseases.

Best regards,

Shaden Kamhawi

co-Editor-in-Chief

Paul Brindley

co-Editor-in-Chief
